# Assessing allocation bias in stratified clinical trials with multi-component endpoints evaluated using the stratified Wei-Lachin test

**Stefanie Schoenen**[1]\*, **Nicole Heussen**[1,2], **Ralf-Dieter Hilgers**[1,2]

1 Institute of Medical Statistics, RWTH Aachen University, Aachen, Germany, 2 Medical School, Sigmund Freud Private University, Vienna, Austria

\* stschoenen@ukaachen.de

## Abstract

**Background:** A common issue in rare disease stratified clinical trials with multi-component endpoints is allocation bias, as they frequently lack blinding. Allocation bias arises when future treatment allocations can be predicted from prior ones, potentially leading to patients with specific characteristics being preferentially assigned to either the treatment or control group. Despite its potential impact, the effect of allocation bias on inference in these trials remain unstudied.

**Methods:** To model biased patient responses, we derived an allocation biasing policy tailored to stratified trials with multi-component endpoints. Using this policy, we assessed the impact of allocation bias by evaluating type I error rates of a stratified version of the Wei-Lachin test, integrating Fleiss's stratified test with the Wei-Lachin test, when allocation bias was present but ignored during inference.

**Results:** Ignoring allocation bias when applying the stratified Wei-Lachin test results in an inflation of the type I error rate, exceeding the 5% significance level. The amount of inflation depends on the number of strata, number of endpoint components and the chosen randomization procedure. Less restrictive randomization procedures, such as the stratified Big Stick Design, exhibited the lowest type I error inflation, while stratified Permuted Block Randomization, results in highest inflation. The inflation of the type I error increases with the number of strata included. An increasing number of independent endpoint components is also associated with higher inflation.

**Conclusion:** Allocation bias threatens the validity of stratified clinical trials with multi-component endpoints evaluated using the stratified Wei-Lachin test and should be mitigated through careful study planning. Ensure the number of patients in each stratum is not smaller than the number of strata, restrict the number of endpoint components to those essential for the study's objectives, and use randomization procedures that allow for some imbalances, such as the Big Stick Design, to reduce allocation predictability.

**Data availability statement:** Codes to reproduce the results of the simulation study are available in the "Assessing allocation bias in stratified clinical trials with multi-component endpoints evaluated using the stratified Wei-Lachin test" repository, https://doi.org/10.6084/m9.figshare.30670883.

**Funding:** RDH is coordinator, NH task lead of RealiseD supported by the Innovative Health Initiative Joint Undertaking (IHI JU) under grant agreement No 101165912. The JU receives support from the European Union's Horizon Europe research and innovation programme and COCIR, EFPIA, Europa Bio, MedTech Europe, and Vaccines Europe. Views and opinions expressed are those of the author(s) only. This publication reflects the author's views. They do not necessarily reflect those of the Innovative Health Initiative Joint Undertaking and its members, who cannot be held responsible for them. The funders had no role in study design, data collection and analysis, decision to publish, or preparation of the manuscript.

**Competing interests:** The authors have declared that no competing interests exist.

## Introduction

Allocation bias arises when subsequent patient assignments can be predicted based on previous allocations. Then, patients with certain characteristics may be preferentially assigned to either the treatment or control group, which systematically violates the structural equality of both groups, even in randomized clinical trials [1]. Berger refers to this as third-order 'selection bias' [2]. Since rare disease (RD) clinical trials are frequently unblinded or single-blinded, allocation bias is a common concern that may compromise the validity of the trial [3].

Due to the wide geographic distribution of patients and multifaceted nature of RDs, RD clinical trials may benefit from design features such as stratification and multi-component endpoints that aggregate several endpoint components into a single score or rating [3–5]. Using multi-component endpoints and stratifying randomization or analysis can reduce heterogeneity, capture treatment effects on multiple facets of the diseases and may improve statistical power [6,7].

The EMA recommends that the potential contribution of bias to the trial inference needs to be evaluated [8]. The impact of allocation bias has been evaluated in clinical trials with multiple endpoints that are assessed individually rather than being aggregated into a score or rating, as well as in stratified trials [9,10]. However, the impact of allocation bias in stratified clinical trials with multi-component endpoints has not yet been studied.

This paper aims to quantify allocation bias in stratified clinical trials with multi-component endpoints analyzed using the stratified Wei-Lachin (WL) test. We will provide design recommendations regarding the number of strata, and endpoints, as well as the randomization procedure, to minimize the potential impact of allocation bias on trial results and increase validity. Our focus will be on small sample studies.

The paper is structured as follows: First, we introduce the notation and present a model for biased patient responses. Based on this framework, we formulate a stratified version of the WL test for the evaluation of stratified clinical trials with multi-component endpoints, integrating the Fleiss stratified test with the WL test [11,12]. To analyze the effects of bias, we derive an allocation bias policy tailored to stratified trials with multi-component endpoints using Blackwell and Hodges 'convergence strategy' [13]. Then, we outline the settings of a simulation study designed to examine the impact of allocation bias on the test decision of the stratified WL test for different trial settings to set up design recommendation for allocation bias mitigating trials. For this simulation study, we derive a formula to calculate the type I error rate of the stratified WL test when allocation bias is present but ignored during inference. Finally, we summarize the findings, discuss potential limitations and highlight opportunities for further methodological improvements.

## Methods

This section provides the foundation for quantifying allocation bias effects in stratified clinical trials with multi-component endpoints. First, we introduce the notation and assumptions needed to formulate a statistical model for the biased responses in such trials. A stratified version of the WL test is then presented, followed by a model for

quantifying allocation bias. Finally, the design of a simulation study is outlined. The approach for quantifying allocation bias is guided by studies that have analyzed allocation bias in other study designs, such as parallel-group single endpoint trials [14] and single endpoint trials with stratified randomization [10].

## Notation and preliminaries

We consider a randomized, two-arm, parallel-group clinical trial stratified by a fixed factor, e.g. centers or geographical regions, into $K$ independent strata, with a multi-component endpoint that consists of $m \geq 2$ not necessarily independent continuous components. Data are collected at a single time point and no interim or adaptive analysis is conducted. The trial compares treatment **E** with control **C**. The responses to treatment **E** and **C**, respectively, regarding the $m$ endpoints are given by the independent, normally distributed response vectors $X_{j,i} \in \mathbb{R}^m$, $j \in \{1, \dots, K\}$, $i \in \{1, \dots, n_j\}$. The total number of patients in stratum $j$ denoted by $n_j = n_{j,E} + n_{j,C}$, consists of $n_{j,E}$ patients in **E** and $n_{j,C}$ patients in **C**. The total sample size including all strata is then given by $N = \sum_{j=1}^{K} n_j$.

Patients are assigned to the treatment or control group using stratified randomization. The allocation of patients to groups **E** and **C** in stratum $j$ is stored in the allocation vector $t_j = (t_{j,1}, \dots, t_{j,n_j})^T$, where $t_{j,i} = 1$ ($t_{j,i} = 0$) if patient $i$ in stratum $j$ is assigned to **E** (**C**). Stratified randomization is achieved by generating independent randomization lists $t_j \in \{0,1\}^{n_j}$ for each strata [15]. We focus on the following restricted randomization procedures (RP), which force balanced allocation between treatment groups within strata [15]:

- Efron's Biased Coin ($EBC(p)$): Patients are assigned using a biased coin toss, favoring the less frequently allocated group with a probability $p \geq 0.5$ [16].
- Big Stick Design ($BSD(a)$): Patients are allocated based on a fair coin toss until a maximum tolerated imbalance $a$ between the group sizes is reached, then the next patient is assigned to the group with fewer allocations [17].
- Maximal Procedure ($MP(a)$): Patients are allocated regarding a randomization list that is uniformly selected from the sequences of $BSD(a)$ [18].
- Random Allocation Rule ($RAR$): Patients are randomized so that half of them are allocated to the treatment group, and the other half to the control group [19].
- Permuted Block Design ($PBR(k)$): Patients are allocated in blocks of length $k$ and RAR is used within each block [20].

Assuming there is no treatment by strata interaction, but rather a potential allocation bias effect, the patient responses can be modeled as:

$$X_{j,i} = \mu_E t_{j,i} + \mu_C (1 - t_{j,i}) + \tau_{j,i} + \epsilon_{j,i}, \tag{1}$$

where $j \in \{1, \dots, K\}$, $i \in \{1, \dots, n_j\}$ and $\epsilon_{j,i} \sim \mathcal{N}_m(0_m, \Sigma)$ with $0_m \in \mathbb{R}^m$ being the m-dimensional zero vector and $\Sigma$ the common but unknown covariance matrix. The vectors $\mu_E \in \mathbb{R}^m$ and $\mu_C \in \mathbb{R}^m$ represent the expected responses of the treatment or control group, respectively, and $\tau_{j,i} \in \mathbb{R}^m$ captures the allocation bias effect affecting the responses of patient $i$ in stratum $j$ regarding the $m$ endpoints.

The trial aims to examine whether the treatment group is superior to the control group regarding the $m$ concordant endpoint components across strata. This leads us to the multivariate one-sided test problem

$$\mu_E - \mu_C = 0_m \quad \text{vs.} \quad \mu_E - \mu_C > 0_m, \tag{2}$$

where ">" refers to the component-wise relation.

## Stratified Wei-Lachin test

Wei and Lachin proposed a summary statistic that uses the cumulative patient responses across the different endpoint components to test the hypotheses (2) [12,21]. Since stratified randomization necessitates a corresponding stratified analysis [22], we need to establish a stratified version of the WL test. Therefore, we combine the WL approach with the approach of Fleiss, who introduced a stratified analysis by using a weighted t-statistic [11]. Combining both approaches yields the weighted test statistic

$$t_{WL} = \frac{d^T \sum_{j=1}^{K} w_j D_j}{\sqrt{\sum_{j=1}^{K} \frac{w_j^2}{w_j^*}} \sqrt{d^T \hat{\Sigma}_p d}} \tag{3}$$

that can be used for a stratified analysis of the multivariate test problem (2). Note that

- $D_j = \overline{X_{j,E}} - \overline{X_{j,C}}$ is the m-dimensional vector of the mean treatment differences of the endpoint components in stratum $j$ with $\overline{X_{j,E}} = \frac{1}{n_{j,E}} \sum_{i=1}^{n_j} X_{j,i} t_{j,i}$ and $\overline{X_{j,C}} = \frac{1}{n_{j,C}} \sum_{i=1}^{n_j} X_{j,i} (1 - t_{j,i})$,
- $d \in \mathbb{R}^m$ is a given vector and in the following chosen as $(1, \dots, 1)^T$ [23],
- $w_j^* = \frac{n_{j,E} n_{j,C}}{n_{j,E} + n_{j,C}}$ is a variance minimizing weight [11],
- $w_j$ is a deterministic weight associated with strata $j$ and in the following chosen as $w_j = w_j^* = \frac{n_{j,E} n_{j,C}}{n_{j,E} + n_{j,C}}$,
- $\hat{\Sigma}_p = \dfrac{\sum_{j=1}^{K} \sum_{i=1}^{n_j} (X_{j,i} - \overline{X_{j,E}})(X_{j,i} - \overline{X_{j,E}})^T t_{j,i} + (X_{j,i} - \overline{X_{j,C}})(X_{j,i} - \overline{X_{j,C}})^T (1 - t_{j,i})}{\sum_{j=1}^{K} n_{j,E} + n_{j,C} - 2}$ is the pooled covariance matrix [23].

## Allocation biasing policy

Inadequate concealment and blinding in clinical trials may allow prediction of upcoming patient allocations based on previous assignments, resulting in allocation bias. The effect of allocation bias on patients' responses is quantified using a biasing policy. In the case of multiple endpoints, Schoenen introduced a biasing policy to quantify allocation bias [9]. Incorporating stratified randomization, Hilgers developed a biasing policy specifically adapted to this framework [10]. Combining these approaches yields a bias model suitable for stratified multivariate trial settings, given by

$$\tau_{j,i} = \begin{cases} -\eta_j, & \text{if} \quad n_{j,E}(i-1) > n_{j,C}(i-1) \\ 0, & \text{if} \quad n_{j,E}(i-1) = n_{j,C}(i-1) \\ \eta_j, & \text{if} \quad n_{j,E}(i-1) < n_{j,C}(i-1) \end{cases} \tag{4}$$

where $\eta_j = (\eta_{j,1}, \cdots, \eta_{j,m})^T \in \mathbb{R}^m$ represents the biasing factors for the $m$ endpoint components in stratum $j$, $n_{j,E}(i-1)$ and $n_{j,C}(i-1)$ denote the number of patients in stratum $j$ allocated to **E** or **C** after $i$–1 assignments. The biasing factor $\eta_{j,l} \geq 0$, $j \in \{1, \cdots, K\}$, $l \in \{1, \cdots, m\}$ can vary across strata and endpoints. When evaluating the ability of clinical trials to mitigate allocation bias in the planning phase, the biasing factors are often chosen from the literature or as a fraction of the effect size [14]. The biasing policy follows the convergence strategy of Blackwell and Hodges [13]. Thus, the next patient in strata $j$ will be allocated to the less frequent allocated group, if it is the treatment group **E**, a better-responding patient ($\eta_j$) is enrolled, or a bad-responding patient ($-\eta_j$) if it is **C**. In the case of a tie between allocations to the two groups, a neutral patient (0) is selected.

## Simulation study

We perform a simulation study following the recommendations of Morris [24] to investigate the impact of allocation bias on the test decisions of the stratified WL test for several clinical scenarios to figure out which design settings mitigate allocation bias best. Therefore, we calculate the misspecified type I error probability (T1E) of the stratified WL test that refers to the T1E computed conditional on an allocation sequence when the stratified WL test is applied to the misspecified model (1) but the allocation bias effects are ignored during inference.

We simulate the misspecified T1Es concerning various RPs in different clinical scenarios, summarized in Table 1. Thereby, we focus on RPs that are most commonly used in stratified clinical trials [22]. The parameters of the RPs are chosen as follows: for EBC, we set $p = \frac{2}{3}$, as recommended in [16]; for BSD, we use $a = 3$, following [25]; for PBR, we choose $k = 4$ as the largest block size that can be applied uniformly across the examined sample sizes $N \in \{12, 32, 64\}$; and for MP we choose $a = 2$ as an example of a small maximum tolerated imbalance. We restrict the simulations to clinical trials with small sample sizes, reflecting our goal of providing design and planning guidance for more valid rare disease trials. We consider the minimum number of endpoint components and strata required for a study to qualify as a stratified clinical trial with multi-component endpoint ($m = 2, K = 2$). Building on this foundation, we then investigate how increasing the number of endpoint components or strata influences the extent of allocation bias effects. Further, we examine endpoints with no, moderate, and strong correlations to reflect a broad range of realistic clinical scenarios. Following prevalent recommendations, we select the allocation bias effect as fraction of the effect size [14]. Thus, we analyze the common allocation bias effect $\eta_{j,l} = \eta$ for all $j \in \{1, \dots, K\}$ and $l \in \{1, \dots, m\}$ selected as fraction $\rho \in \{0.05, 0.1\}$ of the effect size $\Delta_{N,K,m}$ that corresponds to the overall effect size across the multiple outcomes combined in the multi-component endpoint. This allows us to determine if small to moderate allocation bias effects meaningfully impact the T1E of the stratified WL test. The effect size $\Delta_{N,K,m}$ is based on a total sample size of $N$ patients, a 5% significance level, balanced allocation across $K$ strata, $m$ independent endpoint components and 80% power of the stratified WL test. The values of $\Delta_{N,K,m}$ considered in this simulation study are outlined in Table 2.

Further, we investigate the impact of heterogeneous allocation bias effects that vary across endpoint components. Specifically, we focus on clinical trials with $N = 32$ patients, $K \in \{2, 4\}$ strata, and $m = 4$ endpoint components and simulate the misspecified T1Es for allocation bias effects $(\eta_{j,1}, \eta_{j,2}, \eta_{j,3}, \eta_{j,4})$ of $(0.2, 0, 0, 0)\Delta_{32,K,4}$, $(0.05, 0.05, 0.05, 0.05)\Delta_{32,K,4}$ and $(0.17, 0.01, 0.01, 0.01)\Delta_{32,K,4}$ for all $j \in \{1, \dots, K\}$. We chose these heterogeneous allocation bias effects such that the aggregated bias effect across all endpoint components remained at the same overall level. This enabled us to evaluate whether different endpoint-specific allocation bias patterns have different impacts on test decisions.

**Table 1**. Clinical scenarios considered in the simulation study

| Property | Setting |
|---|---|
| Number of endpoint components | $m \in \{2, 4\}$ |
| Variance of endpoint components | $\sigma_l^2 = 1$, for all $l \in \{1, \dots, m\}$ |
| Correlation of endpoint components | $R_\zeta = \begin{pmatrix} 1 & \zeta & \cdots & \zeta \\ \zeta & 1 & \ddots & \vdots \\ \vdots & \ddots & \ddots & \zeta \\ \zeta & \cdots & \zeta & 1 \end{pmatrix} \in \mathbb{R}^{m \times m},\ \zeta \in \{0, 0.5, 0.9\}$ |
| Number of strata | $K \in \{2, 4\}$ |
| Sample size | $N \in \{16, 32, 64\}$ |
| Significance level | 5% |
| Power | 80% |
| Allocation ratio | 1:1 |
| Balancing across strata | Balanced: $K \times \frac{N}{K}$ |
| | Imbalanced (for $K = 2$): $\frac{N}{4} : 3\frac{N}{4}$ |
| Randomization procedure | EBC(0.67), BSD(3), MP(2), RAR, PBR(4) |

**Table 2.** The effect size $\Delta_{N,K,m}$ based on a total sample size of *N* patients, a 5% significance level, balanced allocation across *K* strata, *m* independent endpoint components and 80% power of the stratified WL test

| N | K | $\Delta_{N,K,m}$ | |
|---|---|---|---|
| | | m = 2 | m = 4 |
| 16 | 2 | 0.93 | 0.66 |
| | 4 | 0.97 | 0.69 |
| 32 | 2 | 0.64 | 0.45 |
| | 4 | 0.64 | 0.46 |
| 64 | 2 | 0.45 | 0.32 |
| | 4 | 0.45 | 0.32 |

We also analyze the impact of allocation bias in trials with varying numbers of endpoint components where the allocation bias effect is not defined as fraction of the overall effect size $\Delta_{N,K,m}$, but instead as common allocation bias effect $\eta \in \{0.1, 0.2, 0.3, 0.4, 0.5, 0.6, 0.7, 0.8, 0.9\}$.

The misspecified T1Es are calculated concerning a sample of $n_{sim} = 10{,}000$ randomization lists generated by EBC(0.67), BSD(3), MP(2), RAR and PBR(4). We use violin plots to illustrate these error rates and compute the mean misspecified T1E for aggregation. Choosing $n_{sim} = 10{,}000$ guarantees that the standard error of the mean misspecified T1E across the different randomization lists is in the magnitude of $10^{-5}$.

The simulations are performed using the R-4.4.3 on the RWTH High-Performance Computer Cluster. We utilized the R package randomizeR (v.1.4.2) developed by Uschner et al. [26] to generate randomization lists for various RPs.

## Results

In this section, we present the theoretical results, the derivation of a formula for calculating misspecified T1E for the stratified WL test, and the results of the simulation study that investigates the impact of allocation bias in stratified clinical trials with multi-component endpoints.

### Type I error under misspecification

We assume that the multi-component endpoint responses across strata are biased by our allocation biasing policy (4) and follow the misspecified model (1).

Hilgers [10] demonstrates that the Fleiss test statistic in a stratified single endpoint clinical trial follows a doubly non-central t-distribution under allocation bias, we show that the stratified WL test statistic (3) also follows a doubly non-central t-distribution when allocation bias is present. For the definition and a comprehensive review of the properties of the doubly non-central t-distribution, we refer to Johnson, Kotz, and Balakrishnan [27, p. 213]. For independent patients and independent strata, the test statistic $t_{WL}$ is distributed through

$$t_{WL} \sim t''(N - 2K; \delta, \lambda), \tag{5}$$

where $t''(N - 2K; \delta, \lambda)$ denotes the doubly non-central t-distribution with $N–2K$ degrees of freedom and non-centrality parameters $\delta$ and $\lambda$ defined as

$$\delta = \frac{d^T \sum_{j=1}^{K} w_j(\mu_E - \mu_C + \overline{\tau_{j,E}} - \overline{\tau_{j,C}})}{\sqrt{\sum_{j=1}^{K} \frac{w_j^2}{w_j^*} d^T \Sigma d}} \tag{6}$$

$$\lambda = \frac{\sum_{j=1}^{K} \sum_{i=1}^{n_j} (d^T \tau_{j,i})^2 - n_{j,E}(d^T \overline{\tau_{j,E}})^2 - n_{j,C}(d^T \overline{\tau_{j,C}})^2}{d^T \Sigma d} \tag{7}$$

with $\overline{\tau_{j,E}} = \frac{1}{n_{j,E}} \sum_{i=1}^{n_j} \tau_{j,i} t_{j,i}$ and $\overline{\tau_{j,C}} = \frac{1}{n_{j,C}} \sum_{i=1}^{n_j} \tau_{j,i}(1 - t_{j,i})$. A formal proof of this result is provided in S2 Appendix.

The non-centrality parameters demonstrate that the distribution of the stratified WL test is directly affected by the allocation bias factor. Consequently, ignoring bias effects in the analysis of the trial with the stratified WL test may lead to distorted test decisions. Assuming a significance level $\alpha = 0.05$, the T1E under misspecification, conditioned on a fixed allocation sequence $t$, for the test problem (2) can be computed by

$$P_{T=t}(t_{WL} > t_{N-2K}(1 - \alpha)|H_0) = 1 - F(t_{N-2K}(1 - \alpha); N - 2K, \delta_0, \lambda), \tag{8}$$

where $F(\cdot; N - 2K, \delta, \lambda)$ denotes the distribution function of the doubly non-central t-distribution with $N$–$2K$ degrees of freedom and non-centrality parameters $\delta_0$ and $\lambda$ determined according to (6) with $\mu_E - \mu_C = 0_m$ and (7). Additionally, $t_{N-2K}(1 - \alpha)$ denotes the $(1 - \alpha)$-quantile of the central t-distribution with $N$–$2K$ degrees of freedom.

Using (8), we can examine the impact of allocation bias on the stratified WL test's decisions across various clinical scenarios to select design features that leads to less inflation of the T1E when allocation bias is present.

## Simulation results

As outlined in section "Simulation study", we conducted a simulation study to analyze the impact of allocation bias on the inference of the stratified WL test for various clinical scenarios by computing the misspecified T1Es according to (8). We aim to provide design recommendations for stratified clinical trials with multi-component endpoints to prevent distortion caused by allocation bias.

Fig 1 displays violin plots of misspecified T1Es calculated by (8) using samples of 10,000 randomization lists of different RPs in clinical trials with $N = 32$ patients, $K \in \{2, 4\}$ strata and $m \in \{2, 4\}$ endpoints and allocation bias effects $\eta_{j,l} = \eta$ for all $j \in \{1, \ldots, K\}$ and $l \in \{1, \ldots, m\}$ chosen as proportion $\rho \in \{0.05, 0.1\}$ of the effect size $\Delta_{N,K,m}$. The black dots represent the mean misspecified T1Es of the corresponding simulation scenario. The numerical values of the mean misspecifed T1Es are shown in S3 Appendix. The simulations indicate that ignoring allocation bias when it is present leads to inflation of the T1Es, increasing the likelihood of erroneous conclusions. The violin plots demonstrate that in the presence of allocation bias, the stratified WL test fails to maintain the predefined significance level of 5% (red dashed line). The inflation depends on the chosen RP. Among the RPs analyzed, the RPs associated with the least inflation are BSD(3) and EBC(0.67) and the RP with the highest inflation is PBR(4). We also find that including more strata leads to slightly more inflation of the misspecified T1Es. This effect is more pronounced in small sample trials, as illustrated in Fig 2, which shows the misspecified T1Es in clinical trials with $N \in \{16, 32, 64\}$ patients, $K \in \{2, 4\}$ strata, and $m = 2$ endpoint components under common allocation bias effects of 10% of the effect size $\Delta_{N,k,m}$.

When selecting the allocation bias not based on the overall effect size of the multi-component endpoint, but rather on the effect size of each endpoint component separately, we observe in Fig 3, which illustrate the mean misspecified T1Es for allocation bias effects of $\eta \in \{0, 0.1, 0.2, 0.4, 0.4, 0.5, 0.6, 0.7, 0.8, 0.9, 1\}$ with $\eta_{j,l} = \eta$ for all $j \in \{1, \ldots, K\}$ and $l \in \{1, \ldots, m\}$ in clinical trials with $N = 32$ patients, $K = 2$ strata and $m \in \{2, 4\}$ endpoint components, that including more endpoint components is associated with greater inflation of the mean misspecified T1Es.

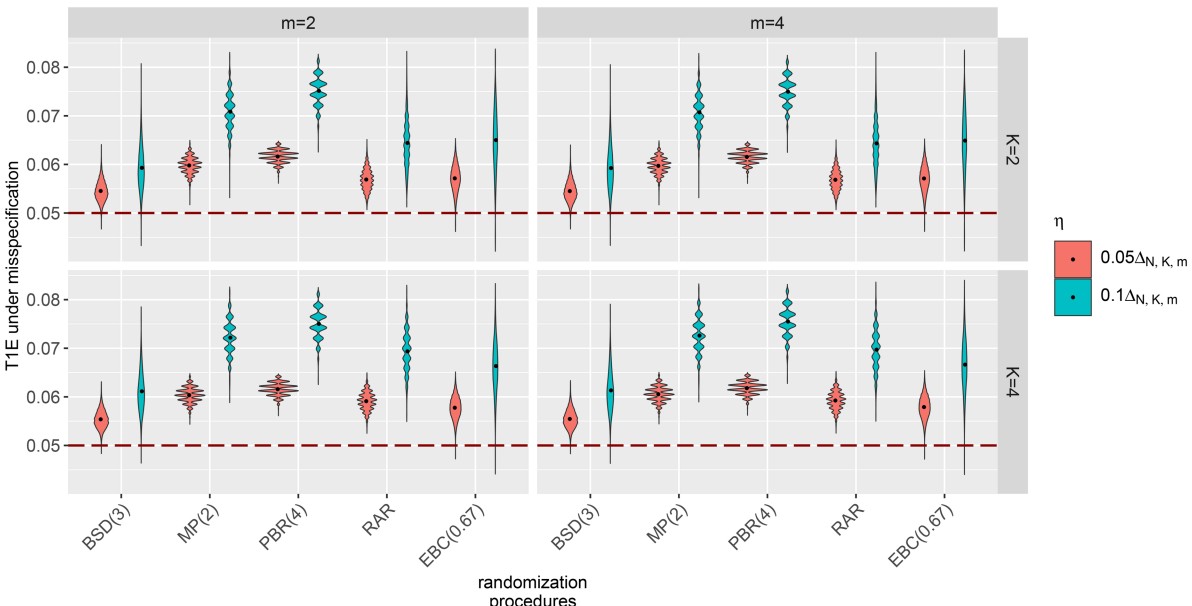

**Fig 1**. **Impact of allocation bias in stratified clinical trials with multi-component endpoints evaluated using the stratified WL test.** The T1Es under misspecification calculated for samples of 10,000 randomization lists generated by different RPs in clinical trials with $N = 32$ patients, $K$ balanced strata, $m$ standard normally distributed uncorrelated endpoints and common allocation bias effect $\eta = \eta_{j,l}$ for all $j \in \{1, \ldots, K\}$ and $l \in \{1, \ldots, m\}$ chosen as proportion $\rho \in \{0.05, 0.1\}$ of the effect sizes $\Delta_{32,2,2} = 0.64$, $\Delta_{32,2,4} = 0.45$, $\Delta_{32,4,2} = 0.64$ and $\Delta_{32,4,4} = 0.46$.

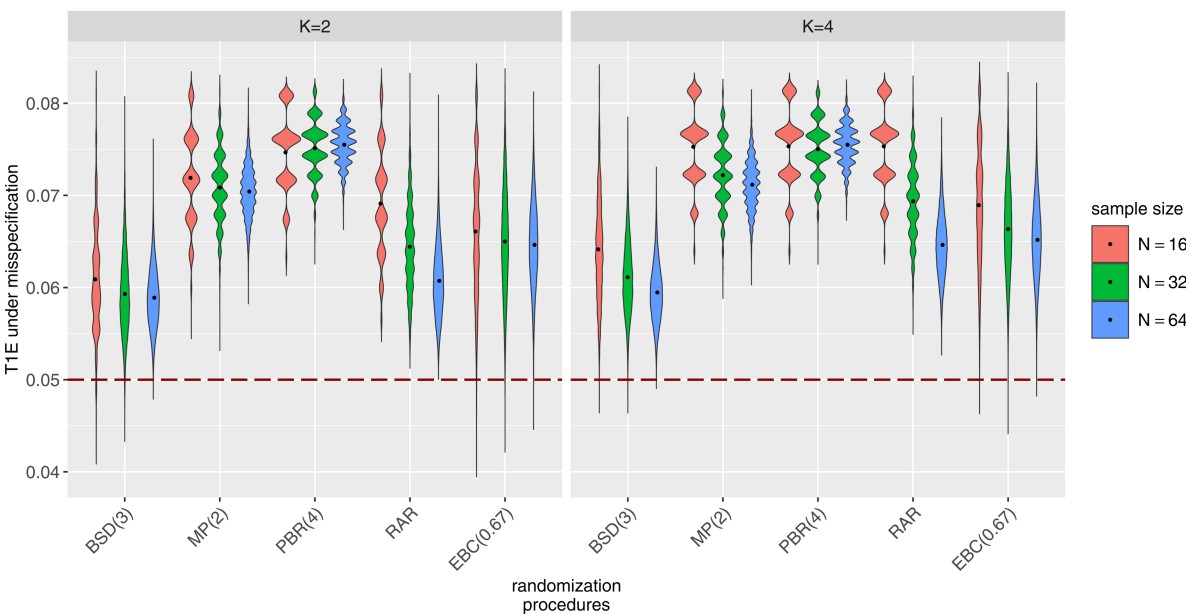

**Fig 2**. **Impact of allocation bias on the test decisions of the stratified WL test depending on the sample size and the number of strata.** T1Es under misspecification calculated for samples of 10,000 randomization lists generated by different RPs in clinical trials with $N \in \{16, 32, 64\}$ patients, $K \in \{2, 4\}$ balanced strata, $m = 2$ standard normally distributed uncorrelated endpoints and common allocation bias effect $\eta = \eta_{j,l} = 0.1\Delta_{N,K,m}$ for all $j \in \{1, \ldots, K\}$ and $l \in \{1, \ldots, m\}$ with $\Delta_{32,2,2} = 0.64$ and $\Delta_{32,4,2} = 0.64$.

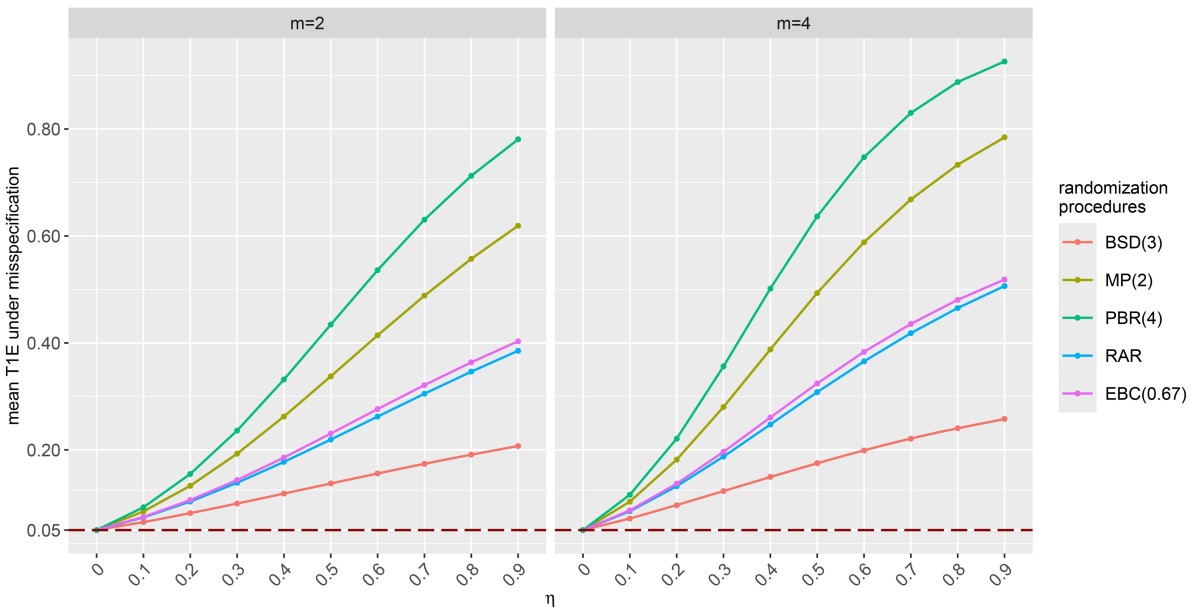

**Fig 3. Impact of allocation bias on the test decisions of the stratified WL test for increasing number of endpoint components aggregated into the multi-component endpoint.** Mean misspecified T1Es calculated for samples of 10,000 randomization lists generated by different RPs in clinical trials with $N = 32$ patients, $K = 2$ balanced strata, $m \in \{2, 4\}$ standard normally distributed uncorrelated endpoints and common allocation bias effect $\eta = \eta_{j,l}$ for all $j \in \{1, \ldots, K\}$ and $l \in \{1, \ldots, m\}$ with $\eta \in \{0, 0.1, 0.2, 0.3, 0.4, 0.5, 0.6, 0.7, 0.8, 0.9\}$.

Previously, we only focused on the case of uncorrelated endpoint components. Fig 4 depicts the misspecified T1Es of a trial with $N = 32$ patients, $K = 2$ strata and $m = 4$ correlated endpoint components for bias effects of $\eta = \eta_{j,l} = 0.1\Delta_{N,k,m}$ for all $j \in \{1, \ldots, K\}$ and $l \in \{1, \ldots, m\}$. The endpoints are correlated by a correlation matrix that follows compound

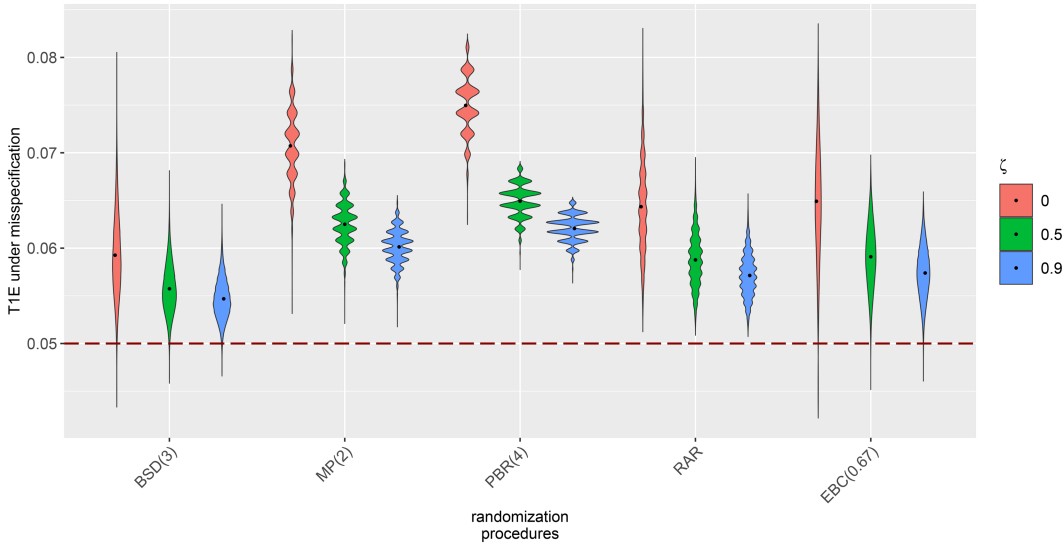

**Fig 4. Impact of allocation bias on the test decisions of the stratified WL test for correlated endpoint components aggregated into the multi-component endpoint.** Misspecified T1Es calculated for samples of 10,000 randomization lists generated by different RPs in clinical trials with $N = 32$ patients, $K = 2$ balanced strata, $m = 4$ correlated normally distributed uncorrelated endpoints and common allocation bias effect $\eta = \eta_{j,l} = 0.1\Delta_{N,K,m}$ for $j \in \{1, \ldots, K\}$ and $l \in \{1, \ldots, m\}$ with $\Delta_{32,2,4} = 0.45$. All endpoint components are equally correlated by the factor $\zeta \in \{0, 0.5, 0.9\}$.

symmetry, in other words, all endpoint components are equally correlated by $\zeta \in \{0, 0.5, 0.9\}$. The figure shows that uncorrelated endpoint component with $\zeta = 0$ builds the worst case scenario regarding the T1E inflation due to allocation bias.

The former simulations only examined allocation bias effects that were equal across endpoint components. We refer to these as common or homogeneous allocation bias effects. When considering heterogeneous allocation bias effects that differ between endpoint components, we find that an increasing sum of endpoint-specific biasing factors leads to greater inflation of the misspecified T1Es. It is irrelevant whether the bias effect is evenly distributed across all endpoint components or is only caused by one endpoint component, both yield the same extent of T1E inflation. Further, we found that the impact of allocation bias in stratified clinical trials with multi-component endpoints depends not on the balancing between strata. Simulation results regarding heterogeneous bias effects as well as for different balancing scenarios between strata are presented in S3 Appendix.

Based on the results of the simulation study, Table 3 outlines recommendations for an allocation bias mitigating trial design in small sample stratified clinical trials with multi-component endpoints evaluated by the stratified WL test.

## Discussion

Allocation bias is an issue that may distort trial outcomes. Its impact on trial results is fostered by design components such as the blinding conditions, the RP used, the number of strata and the number and type of endpoints. Regarding the ICH E9 the potential contribution of bias to inference should be analyzed [8]. The impact of allocation bias is often analyzed only qualitatively, for example using RoB 2.0 [28]. We developed a methodology that quantifies allocation bias in stratified clinical trials with multi-component endpoints. Therefore, a test strategy for these trials is introduced by combining the stratified testing approach of Fleiss with the WL test for multi-component endpoints [11,12] and integrating the allocation biasing policy for multiple endpoints [9] with those for stratified clinical trials [10]. The allocation bias policy modeled allocation bias corresponding to a worst-case scenario. For the evaluation of the impact of allocation bias on the inference of the stratified WL test we determined a formula for the T1E under misspecification, that results when bias is present but ignored during inference. In a simulation study, we quantify the impact of allocation bias on test decisions of the stratified WL test by examining the misspecified T1E across several clinical scenarios. We have shown that in stratified clinical trials with multi-component endpoints evaluated with the stratified WL test, ignoring allocation bias leads to inflation of the T1Es so that the predefined nominal significance level is not maintained. Including more strata strengthens this inflation, especially in small sample trials. Ensuring that the number of patients per stratum is not smaller than the number of strata reduce the risk of distortions due to allocation bias. Balancing across strata exerts only a minor influence on the impact of allocation bias, consistent with the results reported by Hilgers for single endpoint stratified clinical trials [10].

Using multi-component endpoints provides more information about patients and simplifies the classification of patients as good, neutral, or bad responders, as well as the subsequent biased allocation. Increasing the number of endpoint components leads to greater inflation of the misspecified T1E. For minimizing the risk of distortion due to allocation bias it is essential to limit the number of endpoint components to those necessary for the study objective. This is particularly

**Table 3**. **Key lessons learned for mitigating allocation bias in small sample stratified clinical trials with multi-component endpoints that are evaluated using the WL test.**

| Recommendation | Explanation |
|---|---|
| Limit the number of strata | Ensure the number of patients per stratum is at least equal to the total number of strata to reduce allocation bias. |
| Limit the number of endpoint components | Include only essential components in multi-component endpoints to maintain a robust assessment of the treatment effect. |
| Use less restrictive RPs | Less restrictive RPs that allow limited imbalances (e.g., BSD), helping mitigate bias. |

important when developing patient-centered outcome measures (PCOMs) for rare diseases [29,30]. Whether allocation bias arises from a single endpoint component or is distributed across multiple components, the overall inflation of the misspecified T1E increases with the cumulative allocation bias effect. Uncorrelated endpoints represents the worst-case scenario regarding T1E inflation due to allocation bias. Thus, trials at risk of allocation bias must carefully balance the inclusion of additional uncorrelated endpoints to comprehensively characterize treatment effects against the increased risk of T1E inflation. Comparable results were observed when evaluating multiple primary endpoints using the Šidák correction or co-primary endpoints using the all-or-none procedure [4,9]. However, misspecified T1E inflation was less strongly influenced by the number of endpoints considered. When comparing the impact of allocation bias across trials with multi-component endpoints, co-primary endpoints analyzed using the all-or-none procedure, and multiple primary endpoints evaluated using the Šidák correction, trials with multi-component endpoints assessed with the WL test appear most susceptible to T1E inflation due to allocation bias. In contrast, trials with multiple primary endpoints show the least inflation of error rates under allocation bias. It should be noted that the comparability of allocation bias effects across the three endpoint types is naturally limited, as each reflects a different research objective. The aim of studies with multiple primary endpoints is to demonstrate a treatment effect on at least one endpoint, whereas the goal of studies with co-primary endpoints is to show an effect on all endpoints, and the focus of studies with multi-component endpoints is on a newly defined aggregated endpoint [4]. Nevertheless, when each endpoint contributes to the overall bias effect, error inflation is most pronounced when multi-component endpoints are used. This underscores the need for bias-mitigating study designs when trials rely on multi-component endpoints. Simulation results comparing the impact of allocation bias considering these three types of endpoint are presented in S4 Appendix.

The amount of T1E inflation depends also on the chosen RP. We focused on restricted RPs, as they are most common in stratified clinical trials [22] and found that relaxing the final balance of the allocations to the treatment and control group per strata reduce this inflation. The strongest inflation occurs when stratified PBR is used and the least inflation when BSD is implemented. The more restrictive the RP, the greater the inflation. Similar findings have been shown in other studies examining allocation bias in other clinical settings, such as in stratified trials with single endpoints [10] or trials with time-to-event data [31]. In stratified clinical trials the most common RP is stratified PBR, however, as the simulations has been shown, these trials are potentially affected by allocation bias, especially when there is a lack of blinding [22]. Given that the ICH E9 guideline still only refers to block randomization in its randomization section, it is important to inform the scientific community that PBR is outperformed by RPs, such as BSD, in preventing allocation bias [8]. For future trial designs, we recommend stratified randomization procedures that allow for restricted imbalances between the treatment groups, such as the BSD, to reduce the predictability of upcoming assignments. This will reduce the impact of allocation bias on test decisions and improve trial validity. To determine the most appropriate randomization procedure and their parameter settings for a specific study design, we further advise conducting a simulation study during the planning phase that follows the ERDO framework [14].

If the clinical trial is impacted by allocation bias, due to insufficiently trial planning or unpredictable events, a bias-adjusted sensitivity analysis may improve validity of trial results. Therefore, we can use a regression framework including the allocation bias effect, the treatment and the strata as fix effect and test for a bias-adjusted treatment effect using a F-test. Consistent results of the primary analysis using the stratified WL test and the bias-adjusted sensitivity analysis support study results. However, it should be noted that the sensitivity analyses may be underpowered to detect true treatment or bias effects and, therefore, the best way to prevent distortion due to allocation bias is a bias-mitigating study planning. For a more detailed description of the approach for a bias-adjusted sensitivity analysis we refer to S5 Appendix.

One limitation of this work is that the derived methodology only addresses allocation bias as the most prominent type of bias related to the design component of randomization. However, when considering the effects of other biases, the results may change, and other RPs may perform better [32]. In addition, the focus is on the investigation of allocation bias in stratified clinical trials evaluated with the stratified WL test. It should be noted that the WL test is sensitive to differences

in the measurement scale of endpoint components and is not scale invariant. Standardizing endpoint components, as in the Läuter test, provides a scale-invariant alternative [33]. We mainly focus on continuous, multi-component endpoints because the stratified version of the WL test considered here is restricted to this type of endpoint. However, the original version of the stratified WL test [12] can also handle non-normally distributed data including data with missing values or survival data. The impact of allocation bias on the test decision may differ in these more general scenarios. The simulation study focused on analyzing the effects of allocation bias in a few clinical scenarios involving continuous multi-component endpoints, common allocation bias effects across strata, balanced strata, restricted stratified RPs, small sample sizes, and homogeneous weights across strata and endpoints. The simulations demonstrate that design components such as the number of endpoint components, the number of strata, and the randomization procedure influence the impact of allocation bias on the stratified WL test decision. These aspects should be carefully considered during the planning phase of a clinical trial. To determine a bias-mitigating design for a specific clinical scenario, a tailored simulation study reflecting the study's particular settings and evaluating the relevant design components using the outlined framework and R code may be necessary. Because the magnitude of the biasing factor is typically difficult to predict in practice, robust design recommendations should be based on simulation scenarios that consider a range of plausible bias effects rather than a single assumed value.

Further research may extend the methodology to quantify allocation bias to more complex testing procedures, such as the Läuter test, or to test procedures that also can handle non-continuous endpoint types, as the original version of stratified WL test. However, the issue of how to derive the distribution of this test under biased conditions remains unresolved in these cases. To analyze the impact of allocation bias on test decisions of these more advanced test procedures and determine a bias-mitigating trial design, the T1E under misspecification for each randomization list needs to be simulated using a sample of different clinical trials, which becomes computationally demanding. Nevertheless, the proposed allocation bias policy remains applicable even in these more complex settings. Extending the methodology to mixed endpoint types, including binary, time-to-event and continuous data, are necessary for broader applicability, particularly in the context of patient-centered outcome measures [29]. Generalized pairwise comparison (GPC) offer a flexible approach for analyzing multi-component endpoints consisting of several endpoint types [5,34]. The analysis of allocation bias in the context of the GPC statistic and more complex multi-component endpoints is part of future research [34] and require the extension of the biasing policy to binary endpoints, while for the continuous and time-to-event components the existing biasing policies can be used [14,31]. In the future, the selection of a clinical trial design should also consider the potential impact of other types of bias, such as time trend, to ensure that the selected trial design mitigate bias best and yield the most reliable results [14]. The new open science requirements in the updated CONSORT guideline 2025 [35] create future opportunities to evaluate what are reasonable allocation bias effects in clinical trials.

Although, in this work allocation bias is only examined in trials with multi-component endpoints evaluated with the stratified WL test, it provides a framework for enhancing the validity of trials with multi-component endpoints by identifying bias-mitigating design strategies. It is a starting point that can be adapted to more complex scenarios including more sophisticated testing procedures and multi-component endpoints.

## Conclusion

Allocation bias can inflate the risk of erroneous conclusions in stratified clinical trials with multi-component endpoints evaluated using the stratified WL test. Our methodology helps identify trial designs that best mitigate allocation bias. To reduce the risk of biased results, the number of patients per stratum should not be smaller than the total number of strata, the number of endpoint components should be limited to those essential for a comprehensive assessment of the treatment effect, and the applied RP should be less restrictive, allowing limited imbalances such as the BSD. Careful clinical trial planning and the selection of a bias-mitigating trial design are essential for generating credible results and protect against allocation bias.

## Supporting information

**S1 Appendix. Abbrevations and symbols.** Explanation of abbreviations and symbols that has been used in the manuscript.
(PDF)

**S2 Appendix. Distribution of the stratified Wei-Lachin test statistic under allocation bias.** Proof that the stratified WL test statistic is doubly non-central t-distributed under biased conditions.
(PDF)

**S3 Appendix. Additional simulation results on the impact of allocation bias on the stratified Wei-Lachin test decisions.** Simulation results of the misspecified T1Es under different biased clinical scenarios, including unbalanced strata and heterogeneous bias effects across endpoints.
(PDF)

**S4 Appendix. Comparison of the impact of allocation bias using different types of multiple endpoints.** Comparison of the mean misspecified T1E and family-wise error rates for multiple primary endpoints, co-primary endpoints, and multi-component endpoints, evaluated using the Šidák correction, the all-or-none procedure, and the Wei–Lachin test.
(PDF)

**S5 Appendix. Bias-adjusted sensitivity analysis.** Introduction of an approach for bias-adjusted analysis to improve the validity of trials that are potentially affected by allocation bias.
(PDF)

## Acknowledgments

We gratefully acknowledge the computing time provided by the NHR Center NHR4CES at RWTH Aachen University (project number p0024408).

## Author contributions

**Conceptualization:** Stefanie Schoenen, Ralf-Dieter Hilgers.

**Data curation:** Stefanie Schoenen.

**Formal analysis:** Stefanie Schoenen.

**Funding acquisition:** Ralf-Dieter Hilgers.

**Investigation:** Stefanie Schoenen.

**Methodology:** Stefanie Schoenen, Ralf-Dieter Hilgers, Nicole Heussen.

**Project administration:** Ralf-Dieter Hilgers.

**Software:** Stefanie Schoenen.

**Supervision:** Ralf-Dieter Hilgers, Nicole Heussen.

**Validation:** Stefanie Schoenen.

**Visualization:** Stefanie Schoenen.

**Writing – original draft:** Stefanie Schoenen.

**Writing – review & editing:** Stefanie Schoenen, Ralf-Dieter Hilgers, Nicole Heussen.

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
