## [Decision Letter · Decision Letter 0]

30 Oct 2025

PONE-D-25-50444Assessing allocation bias in stratified clinical trials with multi-component endpoints evaluated using the stratified Wei-Lachin testPLOS ONE

Dear Dr. Schoenen,

Thank you for submitting your manuscript to PLOS ONE. After careful consideration, we feel that it has merit but does not fully meet PLOS ONE’s publication criteria as it currently stands. Therefore, we invite you to submit a revised version of the manuscript that addresses the points raised during the review process.

We look forward to receiving your revised manuscript.

Kind regards,

Hui-Juan Cao, Ph.D.

Academic Editor

PLOS ONE

Journal Requirements:

Additional Editor Comments:

This manuscript addresses an important methodological gap in the context of stratified clinical trials with multi-component endpoints, particularly focusing on allocation bias—a critical issue in rare disease trials that often lack blinding. The authors propose a novel allocation biasing policy tailored to such trials and evaluate its impact on the type I error rate of the stratified Wei-Lachin test. The topic is timely, and the methodology appears rigorous. However, several areas require attention before the manuscript can be considered for publication.

The derivation of the allocation biasing policy is a notable strength. However, the assumptions underlying the model (e.g., independence of endpoint components) should be explicitly stated and justified.

The simulation study design is comprehensive, but the rationale for selecting specific parameter ranges (e.g., ρ∈{0.05,0.1} should be elaborated upon.

The finding that the Big Stick Design minimizes type I error inflation is compelling. However, the implications of these findings for practical trial design could be discussed more thoroughly.

The manuscript briefly mentions limitations but does not adequately address how these might affect the generalizability of the results.

Reviewer's Responses to Questions

**Comments to the Author**

1. Is the manuscript technically sound, and do the data support the conclusions?

Reviewer #1: Yes

Reviewer #2: Yes

2. Has the statistical analysis been performed appropriately and rigorously?

Reviewer #1: Yes

Reviewer #2: Yes

3. Have the authors made all data underlying the findings in their manuscript fully available?

Reviewer #1: Yes

Reviewer #2: Yes

4. Is the manuscript presented in an intelligible fashion and written in standard English?

Reviewer #1: Yes

Reviewer #2: Yes

5. Review Comments to the Author

Reviewer #1: 1. Significance and Novelty:

This paper addresses a critical and often overlooked issue in the design and analysis of randomized clinical trials, particularly in the context of rare diseases and patient-centered outcomes. The development of a formal methodology to quantify the impact of allocation bias on the stratified Wei-Lachin test is a significant contribution. The finding that common design elements (number of strata, number of endpoint components, and the choice of randomization procedure) can substantially inflate Type I error rates is both important and practical, providing a clear impetus for improved trial planning.

2. Methodological Rigor and Clarity:

The methodological approach is sound and well-structured. The integration of Fleiss's stratified test with the Wei-Lachin test, combined with a formal allocation biasing policy, creates a robust framework for investigation. The use of simulation to evaluate Type I error inflation across various clinical scenarios is appropriate and convincing. However, the manuscript would be strengthened by a more detailed explanation or a reference for the "doubly non-central t-distribution" of the test statistic under bias, as this is a key technical point that may be unfamiliar to many readers.

3. Limitations and Future Work:

The authors appropriately acknowledge the limitations of their work, such as the focus on continuous, normally distributed endpoints and restricted randomization procedures. The discussion of how their framework could be extended to Generalized Pairwise Comparisons (GPC) is a particular strength, as it directly addresses a modern and flexible analysis method for complex endpoints. A recommended addition would be to briefly suggest potential methods for the proposed "bias-adjusted sensitivity analysis using a regression framework," as this is a valuable recommendation for practitioners but is currently mentioned without detail.

Reviewer #2: Dear authors, first of all, we would like to point out that your topic is interesting and necessary.

A few minor observations: what future research would be needed to continue the study, and what are its limitations?

In the methods section, support the methodology used with a citation.

There are other studies that have been conducted in previous years. In this case, indicate what is new in this section (just answer; do not add to the manuscript).

6. PLOS authors have the option to publish the peer review history of their article (what does this mean?). If published, this will include your full peer review and any attached files.

Reviewer #1: No

Reviewer #2: No

---

## [Author Response · Author response to Decision Letter 1]

27 Nov 2025

Dear Editor and Reviewers,

thank you very much for your time and for the constructive comments that helped to improve the manuscript.

In the document “Response_to_the_Reviewer.pdf”, we provide detailed answers to all your questions and outline clearly how the manuscript has been revised to address each of your suggestions. I hope that the implemented changes meet your expectations and adequately address all of your comments.

Kind regards,

Stefanie Schoenen

---

## [Decision Letter · Decision Letter 1]

2 Jan 2026

Assessing allocation bias in stratified clinical trials with multi-component endpoints evaluated using the stratified Wei-Lachin test

PONE-D-25-50444R1

Dear Dr. Schoenen,

We’re pleased to inform you that your manuscript has been judged scientifically suitable for publication and will be formally accepted for publication once it meets all outstanding technical requirements.

Kind regards,

Hui-Juan Cao, Ph.D.

Academic Editor

PLOS One

Additional Editor Comments (optional):

Reviewers' comments:

Reviewer's Responses to Questions

**Comments to the Author**

1. If the authors have adequately addressed your comments raised in a previous round of review and you feel that this manuscript is now acceptable for publication, you may indicate that here to bypass the “Comments to the Author” section, enter your conflict of interest statement in the “Confidential to Editor” section, and submit your "Accept" recommendation.

Reviewer #1: All comments have been addressed

2. Is the manuscript technically sound, and do the data support the conclusions?

Reviewer #1: Yes

3. Has the statistical analysis been performed appropriately and rigorously?

Reviewer #1: Yes

4. Have the authors made all data underlying the findings in their manuscript fully available?

Reviewer #1: Yes

5. Is the manuscript presented in an intelligible fashion and written in standard English?

Reviewer #1: Yes

6. Review Comments to the Author

Reviewer #1: This is a high-quality methodological study that provides important quantitative evidence and practical design recommendations for an under-researched yet practically significant issue in small-sample clinical trials, such as those for rare diseases: the impact of allocation bias on stratified analysis with multi-component endpoints. The study holds clear clinical and methodological significance. The study is technically rigorous and comprehensive. The statistical analysis in the manuscript is appropriate and rigorous.

Key Strengths:

1. Important Topic: Focuses on real-world risks in unblinded or single-blinded stratified trials.

2. Methodological Innovation: Successfully integrates the frameworks of stratified analysis, multi-component endpoint testing, and allocation bias modeling.

3. Practical Conclusions: Offers specific, actionable design recommendations regarding the number of strata, the number of endpoint components, and the choice of randomization procedure, which are highly valuable for trial planners.

4. Strong Reproducibility: Provides complete code.

Minor Suggestions (for the authors' consideration):

Discussion Section: It may be helpful to briefly mention the limitations of the model used in this study (e.g., the assumption that the biasing factor η is known and constant, whereas in practice its magnitude is difficult to predict precisely) and discuss the potential implications of this for the robustness of the design recommendations.

7. PLOS authors have the option to publish the peer review history of their article (what does this mean?). If published, this will include your full peer review and any attached files.

Reviewer #1: No

---

## [Editor Report · Acceptance letter]

PONE-D-25-50444R1

PLOS One

Dear Dr. Schoenen,

I'm pleased to inform you that your manuscript has been deemed suitable for publication in PLOS One. Congratulations! Your manuscript is now being handed over to our production team.

Kind regards,

on behalf of

Dr. Hui-Juan Cao

Academic Editor

PLOS One